



# Application of holography and automated image processing for laboratory experiments on mass and fall speed of small cloud ice crystals

Maximilian Weitzel[1], Subir K. Mitra[1], Miklós Szakáll[2], Jacob P. Fugal[2,3], and Stephan Borrmann[1,2]

[1]Max Planck Institute for Chemistry, Mainz, Germany
[2]Institute of Atmospheric Physics, University of Mainz, Mainz, Germany
[3]Now at SeeReal Technologies

**Correspondence:** Miklós Szakáll (szakall@uni-mainz.de)

**Abstract.** An ice cloud chamber was developed at the Johannes Gutenberg University of Mainz for generating several thousand data points for mass and sedimentation velocity measurements of ice crystals with sizes less than 150 μm. Ice nucleation was initiated from a cloud of supercooled droplets by local cooling using a liquid nitrogen cold finger. Three-dimensional tracks of ice crystals falling through the slightly supersaturated environment were obtained from the reconstruction of sequential holo-
5 graphic images, automated detection of the crystals in the hologram reconstructions, and particle tracking. Through collection of the crystals and investigation under a microscope before and after melting, crystal mass was determined as a function of size. The experimentally obtained mass versus diameter ($m(D)$) power law relationship resulted in lower masses for small ice crystals than from commonly adopted parameterizations. Thus, they did not support the currently accepted extrapolation of relationships measured for larger crystal sizes. The relationship between Best ($X$) and Reynolds ($Re$) numbers for columnar
crystals was found to be $X = 15.3Re^{1.2}$, which is in general agreement with literature parameterizations.

## 1 Introduction

While the size distributions and number concentrations of ice crystals prevalent in different types of clouds throughout the atmosphere are extensively investigated by airborne in-situ measurements and various remote sensing techniques, knowledge of other microphysical properties of these ice particles remains much more elusive (Baumgardner et al., 2017). Thus, the
15 properties of interest are often parameterized to allow the description of important processes like radiative transfer or the evolution of clouds over time in weather and climate models. The ice water content (IWC) in clouds for example has been the subject of several studies but is often difficult to determine accurately. Alternatively, if combined knowledge of the size distribution of ice particles in a cloud and the mass of each individual crystal is available, the IWC can be inferred indirectly. Cotton et al. (2013) described the ice particle mass using an effective density $\rho_{eff}$, defined as the mass of the particle $m$ divided
by the volume of a sphere with diameter equal to the particle's maximum diameter $D_{max}$. Thus, a crystal's mass is given as

$$m(D) = \frac{\pi}{6}\rho_{eff}D_{max}^3. \tag{1}$$





$\rho_{eff}$ is evidently lower than the density of bulk ice, as it accounts for the complex non-spherical shapes of pristine single crystals and aggregates, as well as inclusions of air inside the crystals in the form of small voids or bubbles. Locatelli and Hobbs (1974) and Mitchell et al. (1990), among others, studied $\rho_{eff}$ through ground-based collection of ice crystals, focusing

on the direct analysis of individual crystals. Other studies (e.g. Heymsfield et al. (2010), Cotton et al. (2013)) made use of aircraft-based in-situ observations, deriving relationships between particle size distributions measured by optical array probes and the IWC determined using other instruments. An alternative description of crystal mass can be given by expressions of the generalized form $m(D) = aD^b$, where $a$ and $b$ are empirically derived parameters and $D$ is a representation of the crystal's dimension. With such a relationship, a dependency of $\rho_{eff}$ on particle size is implied, an assumption that is also supported by

theoretical work (Westbrook, 2007).

Another unresolved key parameter in cloud microphysics is the sedimentation velocity of ice crystals of sizes below $150\,\mu m$. Understanding the transport of mass and particle numbers within clouds is essential for accurately modeling many atmospheric processes, such as the formation of precipitation (Heymsfield et al., 2007) and transport and vertical redistribution processes such as denitrification (Molleker et al., 2014). Generally, the terminal velocity of a falling particle is attained if the gravitational

force $F_g$ is equal to the drag force $F_D$ acting on the particle. The drag force experienced by a falling particle can be expressed using a dimensionless drag coefficient $C_d$ as follows:

$$F_d = \frac{1}{2}\rho_a v^2 A C_d, \tag{2}$$

with a crystal falling through air with density $\rho_a$ with velocity $v$ while the area of the crystal projected normal to the fall motion is $A$. When equating Eq. (2) with the gravitational force $F_g = mg$, one obtains an expression for the sedimentation velocity as

$$v = \sqrt{\left(\frac{2mg}{\rho_a A C_d}\right)}. \tag{3}$$

In addition to $A$, the fall velocity evidently also depends (amongst others) on the mass $m$ of the crystal, as well as $C_d$. The latter is a function of the Reynolds number $Re$, which represents the ratio between inertial and viscous forces that govern a particle's motion through the air and can be written as

$$Re = \frac{\rho_a v D}{\eta}, \tag{4}$$

where $\eta$ is the dynamic viscosity of air. The Best number $X$ (Davies, 1945) has been frequently used to elegantly describe fall velocity as a function of the other relevant properties ($m$, $A$, $D$). It is defined as:

$$X = C_d Re^2 = \frac{\rho_a}{\eta^2}\frac{2mgD^2}{A}. \tag{5}$$

As $X$ itself is independent of the fall velocity $v$, relating it to the Reynolds number (which is a function of $v$, but independent of all other particle properties) yields a representative estimation for the particle sedimentation velocity from $m$, $D$ and $A$

(Heymsfield et al., 2010; Mitchell, 1996). This approach proves useful if all of these properties are known or characterizable through approximations and parameterizations.



A different approach for this problem, which was initially described by Hubbard and Douglas (1993), has been adapted by Westbrook (2007). It involves calculating the fall velocity of crystals from the Stokes solution for a falling object with the hydrodynamic radius $R_{hyd}$. While $R_{hyd} = R$ for spherical objects, a suitable description that accounts for the different flow characteristics around the falling object is required for other crystal shapes. If $R_{hyd}$ is known, the fall velocity $v$ can be calculated as

$$v = \frac{g}{6\pi\eta}\frac{m}{R_{hyd}}, \tag{6}$$

which is valid for small Reynolds numbers ($Re \ll 1$, i.e. $R_{hyd} \lesssim 10$ µm) where the flow is dominated by viscous forces.

For both mass and fall velocity, the amount of usable data in the literature is particularly sparse for ice crystal sizes smaller than 150 µm. Currently used parameterizations are often extrapolated from measurements of particles with significantly larger sizes and assumed to also be valid for those small particles. For ice crystal mass in particular, some studies assumed crystals smaller than a certain threshold to have the same mass as a spherical object with the density of bulk ice. Hence, the present study focuses on decreasing the uncertainties in the characterizations of ice crystals in the size range smaller than 150 µm by creating a data set containing the properties of several thousand small ice particles. For this, automated object detection techniques were developed and applied to images and holograms recorded by an experimental setup designed specifically for the purpose of investigating small cloud ice particles. In Sect. 2, the ice cloud chamber that was used for the generation and analysis of the particles in a laboratory is described. Sect. 3 contains a description of the instrumentation and methods utilized for the determination of ice crystal mass and fall velocity. The results obtained from the conducted experiments are discussed in Sect. 4, and a summary and conclusion follow in Sect. 5.

## 2   Ice cloud chamber

An ice cloud chamber (ICC) was developed for the measurement of ice crystal sedimentation velocity through particle tracking in a three-dimensional volume, supplemented with the determination of particle mass through microscopic analysis of their melting product. In the ICC (Fig. 1), which was placed in the walk-in cold room of the Mainz vertical wind tunnel laboratory, locally-produced ice crystals in the size range smaller than 150 µm can be investigated. The main part of the ICC is constructed inside the cold room and has a cylindrical shape spanning 3 m in height and 60 cm in diameter. Air circulation is induced by a fan in a secondary channel connecting the bottom of the chamber to the top (Label 2 in Fig. 1). In order to create a cloud in the chamber volume, this circulation is supplied with droplets generated by an ultrasonic nebulizer (Label 1 in Fig. 1). Once a sufficiently stable cloud has formed, the circulation is stopped, and ice particle nucleation is triggered at the top of the chamber. A hollow copper protruding into the chamber (Label 3 in Fig. 1) is filled with liquid nitrogen, inducing temperatures below 195 K, hence cold enough to trigger homogeneous freezing of the present droplets in the immediate vicinity of the rod. The newly formed crystals grow in the supersaturated environment maintained by the evaporation of liquid droplets while sedimenting towards the bottom of the chamber. The measurement section is mounted to the lowest part of the chamber and connected through an outlet. Particle fall velocity was measured by means of an in-house developed holographic instrument



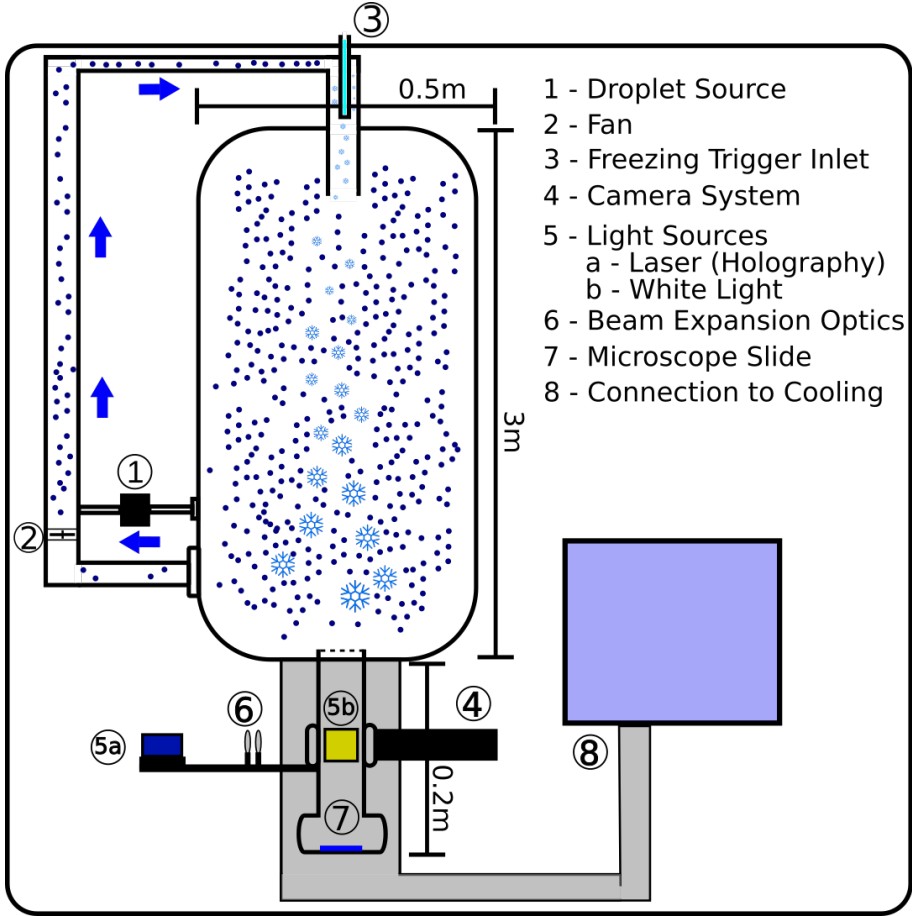

**Figure 1.** Schematic (not to scale) of the ICC from a side view. Droplets are generated and introduced into the chamber from an ultrasonic nebulizer (1) and mixed throughout the chamber through circulation created by a fan (2). After the desired cloud conditions are reached, freezing can be triggered in the top region using a cold finger (3). The measurement section (4-7), where mass and fall velocity measurements are conducted, is suspended below the chamber and ventilated with air from a cooling unit (8) to improve static stability.

(see Sect. 3.1) which is positioned in a way that aligns its optical path through two windows in both of the side walls of the
measurement section. A collector containing a microscope slide positioned in the center of a lid closed off the measurement section at the bottom. This collector was employed to catch the falling crystals for subsequent analysis using a digital camera mounted on a microscope.



# 3 Methodology

## 3.1 Sedimentation velocity

Figure 2 shows the in-house developed "Holographic Imaging and Velocimetry Instrument for Small Cloud Ice" (HIVIS) used
for particle tracking in a sketch **(a)** and a photograph taken from the side **(b)**. HIVIS is an implementation of the classic optical
setup for in-line holography (Silverman et al., 1964; Borrmann et al., 1993; Raupach et al., 2006) as applied to in-situ cloud

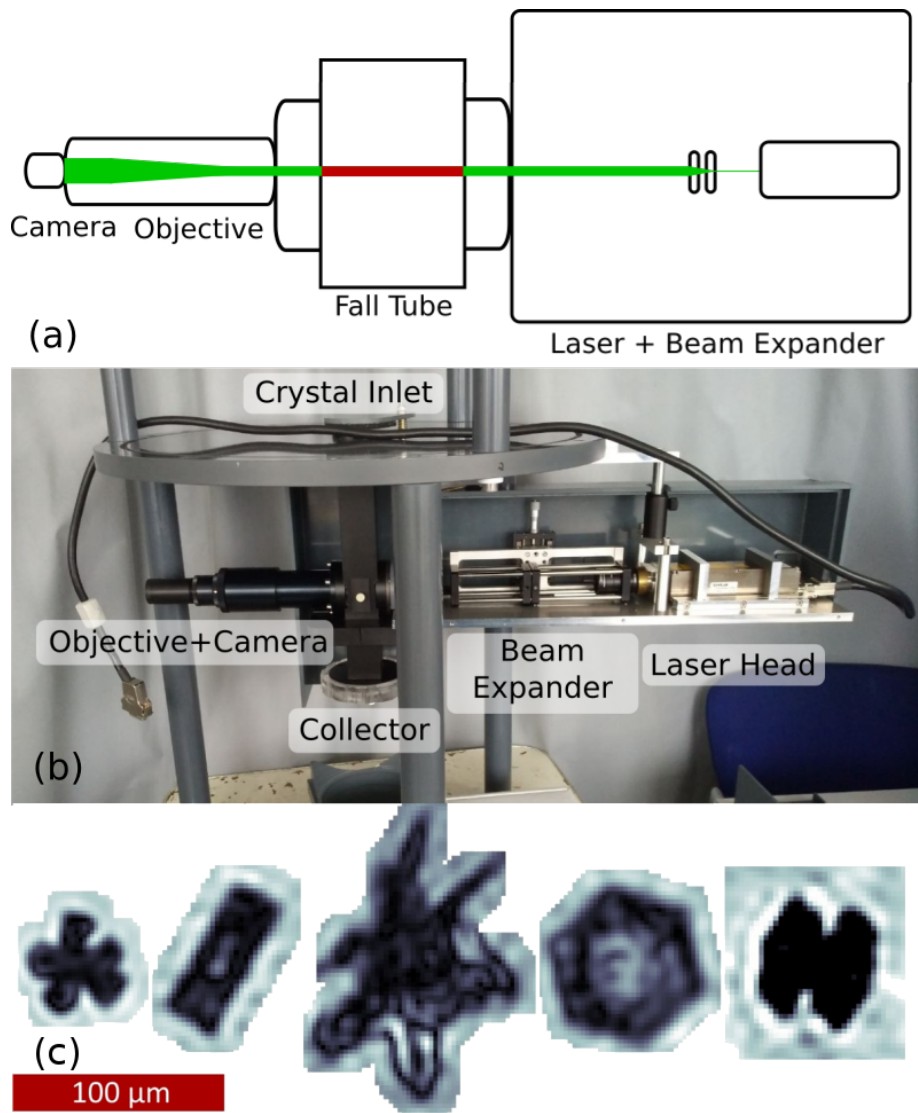

**Figure 2.** The HIVIS instrument used for holographic imaging. **(a):** Sketch of top-down view (sample volume in red), **(b):** photograph taken
from the side. **(c):** sample reconstructed images of ice crystals recorded by the HIVIS instrument.



measurements. The instrument's camera sensor is illuminated by an expanded and collimated beam emitted by a Nd:YAG laser
with a wavelength of 532 nm. The hologram plane created and utilized by this setup has an area of $A_{sample} = 6.2 \times 4.9$ mm$^2$.

Combined with the reconstruction depth of 4 mm, this leads to a sample volume of $4.86$ cm$^3$. The crystals falling through this
sample volume create scattered waves which interfere with the remaining undisturbed part of the laser beam (reference wave).
The interference pattern (the hologram) is recorded by the camera, hence allowing the numerical reconstruction of an in-focus
image of the original particle (Fugal et al., 2004). The camera records about 53 frames per second, yielding at least 3 and up to
10 recordings of crystals during their passage through the sample volume where they fall with a typical velocity of 20 to 100

100  mm/s.

### 3.1.1  Object detection

To prepare for the extraction of data from the recorded holograms, most of the background pattern and speckle noise created by
dirt on the optical surfaces between camera and laser were removed during preprocessing and reconstruction (Fugal et al., 2009;
Schlenczek, 2018). For this, a software filter was applied that divides every pixel's intensity in the hologram recorded at time

$t = t_0$ by a value that represents the median of intensities $\bar{I}_{slice} = \frac{1}{N} \sum_{n=-N/2}^{N/2} I(t_0 + n)$ of this pixel in a set of holograms
recorded shortly before and after $t_0$. The reconstruction for each hologram was then calculated following the convolution
method described in Fugal et al. (2004), resulting in a stack of 2D images with a spatial resolution of $dz = 100$ μm (with $z$
representing the spatial axis along the optical path of the laser) throughout the measurement section for each hologram. An
object detection algorithm was applied which determined the position of the detected objects in three dimensions and their

in-focus images through analysis of several image parameters deduced from both the intensity and phase reconstructions (see
Sect. 5 in Fugal et al. (2009)).

A classification model based on decision trees was created and applied to filter out speckle noise from detections of actual
particles and to separate among different crystal habits. First, a training data set was generated by the operator classifying a
set of several hundred crystal images into one of the following different categories: artifact (disturbance in the reconstructed

image generated by noise), irregular, dendritic, columnar and plate-like. Next, different particle properties were calculated
from the intensity and phase images of each detected object. These properties included simple shape parameters (e.g. axis
lengths of enclosing ellipse, total particle area), derived context information about amplitude and phase, and spatial position
(e.g. distance to image center) to account for image inhomogeneity. From this set of classification data, a decision tree was
created algorithmically in a way that splits the source set of classified objects into different subsets using a binary splitting

criterion that optimizes the split at each node (Breiman et al., 2017). This was done to infer the class membership of the entire
data set from the training subset and the corresponding binarization patterns. The parameters of each object were investigated
following the tree from top to bottom, leading to an unambiguous path which lead to an endpoint representing a class. For
validation purposes, the automated classifier that was generated using this method was applied to a test set of detections and
compared to labels created by the operator, yielding an agreement of over $85\%$.



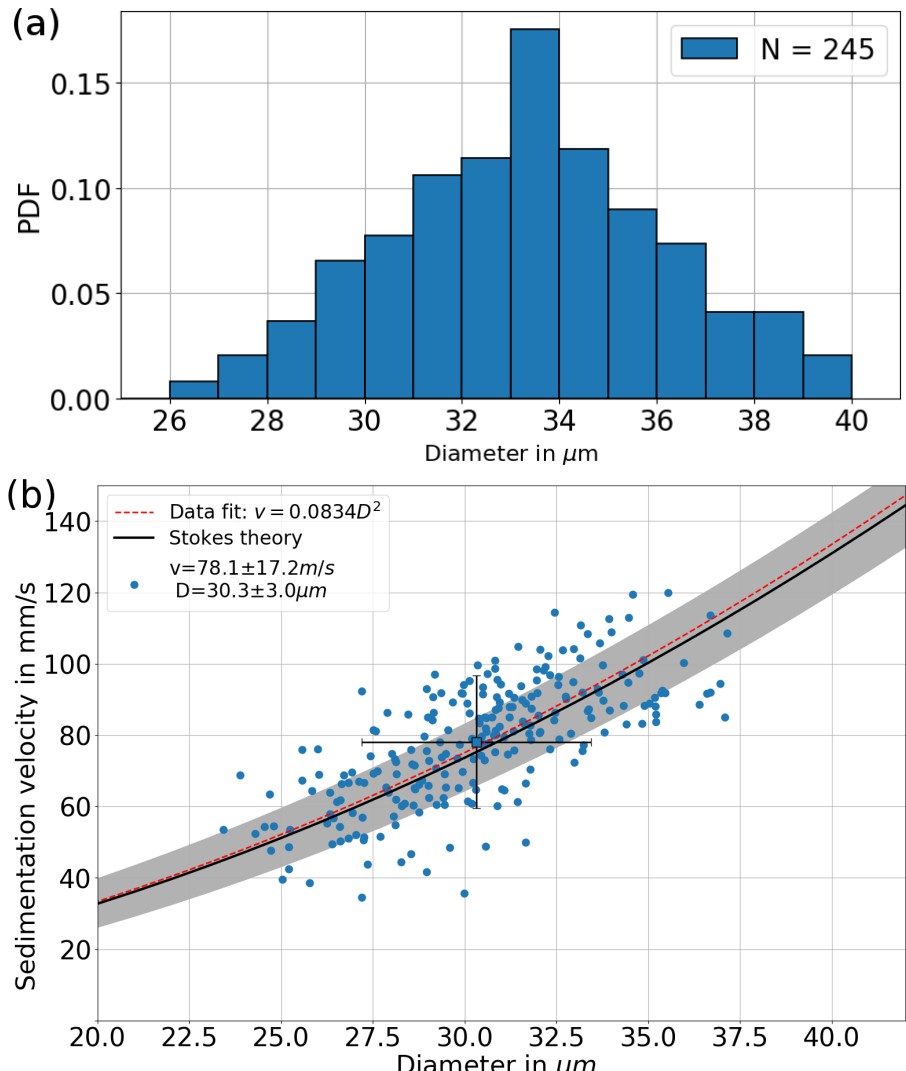

**Figure 3. (a)**: Distribution of measured calibration sphere diameters before size corrections. **(b)**: Fall velocity as a function of (size-corrected) calibration sphere diameter for single calibration measurement fall tracks, quadratic fit as red dashed curve. The black curve shows the velocity expected from Stokes' law as a function of glass sphere diameter for $\rho_g = 2500 \text{ kg m}^{-3}$ with uncertainty as gray shading. The black marker shows mean and standard deviations of the measurement data.

### 3.1.2 Particle tracking

The sedimentation velocity of the falling particles has been determined by tracking their position throughout the three-dimensional sample volume (in the vicinity of the area labeled "5b" in Fig. 1). For each ice crystal object with size $D$ that was detected in the hologram at $t = t_0$, a position $x_{pred}$ in the hologram at $t_1 = t_0 + \Delta t$ is predicted using an estimated fall





velocity $v_{est}$ calculated from the Stokes solution for a sphere with diameter $D$, following

$$130 \quad v = \frac{g}{3\pi\eta}\frac{m}{D}, \tag{7}$$

with $\eta$ being the dynamic viscosity of air and m the crystal mass. If a crystal with similar properties (habit and size) was found close to this predicted position, the actual velocity was calculated from the particle's actual position at $t_1$ and the time step $\Delta t$. Using a leniency distance $L$, crystals are accepted as part of the fall track if their position $x_1$ lies within the region in space defined by $|x_{pred} - x_1| \leq L$. Crystals were tracked through up to 10 holograms this way, and a mean velocity was calculated
from each time step (see Sect. 4.3).

Calibration glass beads were used to conduct reference measurements of particle size and fall velocity for the particle tracking setup. "Dry Soda Lime Glass Microspheres" fabricated by Duke Standards (Fremont, CA, USA), the diameter of which was given by the manufacturer to be $29.5 \pm 1.0\,\mu$m, were recorded while passing through the sample volume. The observed particle sizes, shown in Fig. 3a, showed that a sizing correction had to be applied to the determined particle sizes,
which is common for holographic particle imaging (Lu et al., 2012). The particle size given by the manufacturer was confirmed by measuring the beads under a microscope, as well as by applying the sizing method using a sign-matched filter proposed by Lu et al. (2012) to the recorded holograms. Thus, all particle sizes determined using the approach described in Sect. 3.1.1 were corrected by subtracting a bias of 3 μm. The velocities and corrected diameters determined for 245 calibration beads are shown in Fig. 3b. The velocity calculated for spheres with a given diameter using Eq. (7) is plotted in black, with the uncertainty
from density and size deviations ($\Delta\rho_g = \pm\ 100\,$kg m$^{-3}$, $\Delta D = 2.5\,\mu$m) as gray shading. The measured values are found in the vicinity of the theoretical curve, thus confirming the validity of the method.

### 3.1.3   Fall streak analysis

In a validation experiment, the velocities measured with particle tracking were compared with measurements obtained with a different, independent method. This approach used prolonged camera exposure to obtain a continuous recording of the moving
ice particles' positions over an extended period of time. A fall streak effect with length $s_{str}$ was created in the recorded images for each falling crystal (see Fig. 11a). The projection of the crystal's mean velocity onto the focal plane was then calculated via $v_{fall} = s_{str}/T_{exp}$, with the exposure time $T_{exp}$. The inherent size of the ice crystals, which was in the order of $1\%$ of $s_{str}$, and thus negligible, was ignored in the streak length analysis. The vertical extent of each image was 24 mm. Combined with a camera exposure time of $T_{exp} = 85$ ms, the full length of fall streaks from crystals falling at up to 140 mm/s could be captured
in each recording. As the contrast between the bright streaks created by falling crystals and the dark background was strong, it was possible to use a thresholding method for the automation of streak length measurements, yielding a velocity distribution for each recording. More detailed elaborations on the automated detection of objects in images using thresholding and other techniques are given in Sect. 3.2 and the Supplement.





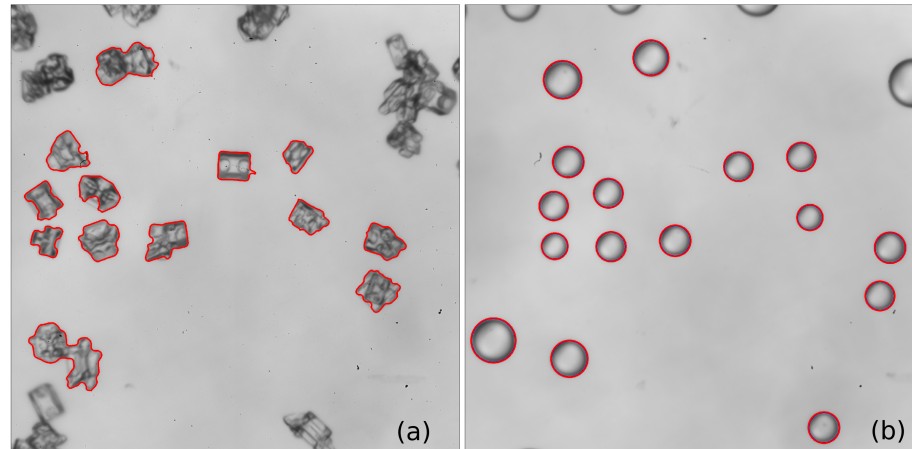

**Figure 4.** Sample images of ice crystals collected on a microscope slide before and after melting. The automatically detected contours (from k-means clustering segmentation (Pedregosa et al., 2011) in the crystal image and from Hough circle detection (Hough and Paul, 1962) in the droplet image) are added in red. Contours which intersect the image frame borders are discarded.

### 3.1.4 Evaluation of residual turbulence

Various steps have been taken in order to suppress any source of turbulence in the fall section, because a calm environment is required to obtain meaningful and unbiased results for the conducted fall speed measurements. Thermal insulation of the sample volume from the light sources required for fall speed measurements and air-tight sealing of the fall section relative to the surrounding cold room were ensured. Further, an air flow from the cooling unit of the ICC directly past the measurement section was created during the experimental process. This ventilation ensured that the fall section containing the measurement

region is the coldest area of the cloud chamber volume, creating a statically stable region within the velocimetry sample volume to inhibit any turbulence potentially disturbing the crystals' falling motion. To verify that the remaining turbulence in the fall section is negligible, test experiments were conducted in which the droplet motion within the sample volume was recorded by a camera. The recorded video was then analyzed and, using tracking of individual droplets, the remaining drift velocity in the improved setup were estimated. The velocity of the weak random turbulent motion of droplets was estimated to be around

$5 \, \mathrm{mm \, s^{-1}}$.

### 3.2 Ice crystal mass

In order to relate the mass of individual ice crystals to a representative particle size, a microscopic imaging method was used. The crystals moving through the fall section (see bottom region in Fig. 1) were collected underneath the chamber on a glass slide treated with a hydrophobic silane. The glass slide was then extracted from the cloud chamber and its surface

was covered with a millimeter-thick layer of oil to prevent sublimation of ice. Next, the coated slide was viewed and scanned under the magnification of a microscope, yielding images of several crystals in each picture. To deduce size information from



| Binarization method | $\overline{|\Delta D_{sec}|}$ | $\overline{|\Delta D_{ae}|}$ |
|---|---|---|
| Global Threshold | 1.4 μm | 1.2 μm |
| Adaptive Threshold | 1.2 μm | 1.7 μm |
| k-means clustering | 1.1 μm | 1.1 μm |
| Canny Edge Detection | 1.5 μm | 2.0 μm |

**Table 1.** Mean error of ice crystal sizing relative to operator-labeled image for different binarization methods in a sample image. $\overline{|\Delta D_{sec}|}$ for diameter of smallest enclosing circle, $\overline{|\Delta D_{ae}|}$ for area equivalent diameter.

these microscope images, we have developed an automated image processing software which utilizes various object detection approaches to accurately trace the crystal edge contours. In addition to global and local grayscale thresholding, Canny edge detection (Canny, 1986) and k-means clustering (Pedregosa et al., 2011) were used to create several binarized representations of each image. From these binary images, the contour tracing approach developed by Suzuki and Abe (1985) was used to create object contours from which characteristic size parameters were obtained. A more thorough elaboration on the segmentation and contour tracing methods can be found in the Supplement.

The accuracy of the particle sizes obtained from the different binarization methods (see Table 1) was evaluated by creating a reference sizing and determining the deviation between particle sizes obtained from the binarized particle representations and the reference sizing. For this, the crystal edges in a sample image have been traced in zoomed-in views of the crystals by an operator. To compare the particle sizes obtained by these reference contours, two size parameters were evaluated: the diameter of the smallest enclosing circle around a contour, $D_{sec}$, and the area-equivalent diameter, $D_{ae}$. The sizing errors of each segmentation method with respect to these parameters were determined by applying them on a sample image containing 12 single crystals (Table 1). Obviously, the sizing error introduced by all methods was smaller than 2 μm, whereas the machine learning-based k-means clustering method provided the best agreement to the shapes determined by the operator. Similar results were observed for other images, with k-means yielding the most accurate results in most cases.

After completion of the crystal image acquisition, the microscope slide was exposed to a heating lamp, which let the ice crystals melt within a few minutes. Subsequently, a second image containing the resulting melted drops was recorded and the droplets' diameters were determined using the circle Hough Transform (CHT) algorithm (Hough and Paul, 1962). Due to the hydrophobic characteristics of the glass surface and the low density of the oil used for coating, the drops formed this way have an approximately spherical shape, allowing for a simple calculation of the water mass contained in each individual ice crystal (Fig. 4b).

Special caution had to be exerted when interpreting drop image data, as the coagulation of multiple melting crystals into a single drop had been observed on several occasions. To prevent this effect from creating a bias in measurement data, affected mass-dimension pairs were removed after the automated image analysis through manual post-processing.



### 3.3 Particle size

As summarized by Wu and McFarquhar (2016), the size of ice crystals is described in a variety of different ways throughout the literature, and an appropriate interpretation is required when comparing size data from different sources. For analysis of the microscope images in this study with the goal of determining particle size, the diameter $D_{sec}$ of the smallest enclosing circle around the detected crystal contour, and the area equivalent diameter $D_{ae}$ were determined and used for deriving the $m(D)$ relationships. For the velocity measurements, the recorded particle images in the described holography setup are 2D projections of the crystals during their fall. The length of the major axis of an ellipse fitted to the particle's contour, $D_{maj}$, was used as the parameter representing particle size.

### 4 Results and discussion

Ice crystal properties were determined by analyzing the images and holograms obtained in a total of 18 experiments conducted in the ICC. In order to produce ice crystals of different habits, the conditions within the chamber during particle growth were varied between experimental runs. The chamber temperature was set to values between -8 and $-16\,^\circ$ and monitored continuously with a thermocouple sensor. Additionally, ice crystal growth is determined by the available water vapor inside of the ICC, which could be influenced indirectly by adjusting the rate and duration of droplet supply into the chamber volume.

### 4.1 Cloud characterization

In order to characterize the thermodynamic conditions of the ICC during typical measurement conditions, the liquid water content (LWC) of the chamber air was determined. For this, the dew point temperature $T_{d,dry}$ of chamber air was determined before an experiment cycle (dry conditions) by sampling chamber air isokinetically into a dew point hygrometer (MBW Calibration Ltd., Wettingen, Switzerland, DP3-D/SH) placed outside the cold room. Afterwards, a cloud of liquid droplets was generated as usual for an experimental run, and chamber air containing droplets was sampled and lead to the hygrometer. In order to evaporate the droplets within the sampled air, the walls of the tube from the chamber towards the hygrometer were heated, inducing an increase in temperature within the tube itself. As relative humidity was thus reduced below saturation, the droplets flowing through the tube evaporated before the sampled air mass reached the dew point hygrometer. The increase in absolute humidity $\Delta q$ within the chamber between dry and cloud-filled conditions is given in Table 2, and it was determined from the measured dew point temperatures and saturation vapor pressures:

$$LWC = \Delta q = \frac{e_{s,cloud}}{R_v T_{d,cloud}} - \frac{e_{s,dry}}{R_v T_{d,dry}}, \tag{8}$$

where $e_{s,cloud}$, $e_{s,dry}$, $T_{cloud}$ and $T_{dry}$ are the saturation vapor pressures and temperatures during cloud and dry conditions and $R_v = 461.4\,\mathrm{J\,kg^{-1}\,K^{-1}}$ is the individual gas constant for water vapor. The saturation vapor pressures were determined using the Magnus approximation to the Clausius-Clapeyron equation (Alduchov and Eskridge, 1996). The increase in dew point temperature of 7.5 K (see Tab. 2) corresponds to a LWC of $1.61\pm0.22\,\mathrm{gm^{-3}}$ within the ICC during typical cloud conditions before nucleation was triggered. This value is similar to observations within typical atmospheric cumulus clouds (Bower and





Choularton, 1988). In a separate experiment using the holography setup described in Sect. 4.3, the droplet size distribution in the fully-formed cloud was determined to have its mode at about $10\,\mu\mathrm{m}$. When combining the determined mean droplet size and LWC, the number concentration of droplets within the ICC cloud can be calculated to be approximately $2500\,\mathrm{cm}^{-3}$.

## 235    4.2    Mass measurements

A total of 1207 pairs of ice crystals and melted droplets were obtained from microscope imaging and the melting technique, with crystal area equivalent diameters between 15 and $145\,\mu\mathrm{m}$. The majority of ice crystals ($\approx 68\%$) showed irregular crystal growth, with complex angular shapes being more frequent than rounded shapes. For pristine crystals, a dependence of growth habit on the thermodynamic conditions was observed. The most frequent pristine shape was columns ($\approx 20\%$), followed by 240   aggregates of pristine and irregular crystals ($\approx 7\%$), and dendrites ($\approx 4\%$). Capped columns, bullet rosettes, and plate crystals were all observed with a fraction of $1\%$ or less.

Figure 5 shows the mass of ice crystals as a function of their size. The blue crosses are data points of the area-equivalent diameter $D_{ae}$ of crystals obtained from the experiments described in Sect. 3, with the blue solid line representing the best power law fit to this data. The solid orange line represents the best fit to data obtained from the experiments in the present study 245   if crystal size is interpreted as the diameter of the smallest enclosing circle around the crystal contour determined by automated object detection ($D_{sec}$). Also added are power law relationships of Cotton et al. (2013), Mitchell et al. (2010), Mitchell et al. (1990), Heymsfield et al. (2010) and Brown and Francis (1995) for comparison. It can be seen that the ice particle masses predicted by most of the parameterizations from the literature are higher than those observed in the present study. An exception is the relation given by Mitchell et al. (2010), which shows good agreement with our parameterizations up to around $100\,\mu\mathrm{m}$.

## 250    4.3    Sedimentation velocity measurements

In Fig. 6, the measurements of ice crystal sedimentation velocity and size are shown for eight experiments conducted in the ICC. Following the varying thermodynamical conditions, different distributions of observed crystal habits were present during each experiment. As expected, a large spread was found in the observed fall velocities, which ranged from a few $\mathrm{mm\,s^{-1}}$ to $120\,\mathrm{mm\,s^{-1}}$.

|  | $T_d$ [K] | $e_s$ [hPa] | q [gm$^{-3}$] |
|---|---|---|---|
| Dry conditions | 262.0±0.5 | 2.66±0.11 | 2.20±0.08 |
| Cloud conditions | 269.5±0.5 | 4.73±0.18 | 3.81±0.14 |

**Table 2.** Dew point temperature and deduced humidity measures for liquid water content measurements. The increase in dew point temperature was caused by the evaporation of droplets on their way from the ICC to the dew point hygrometer. $e_s$ was calculated from the Magnus approximation to the Clausius-Clapeyron equation, $q$ from Eq. (8).



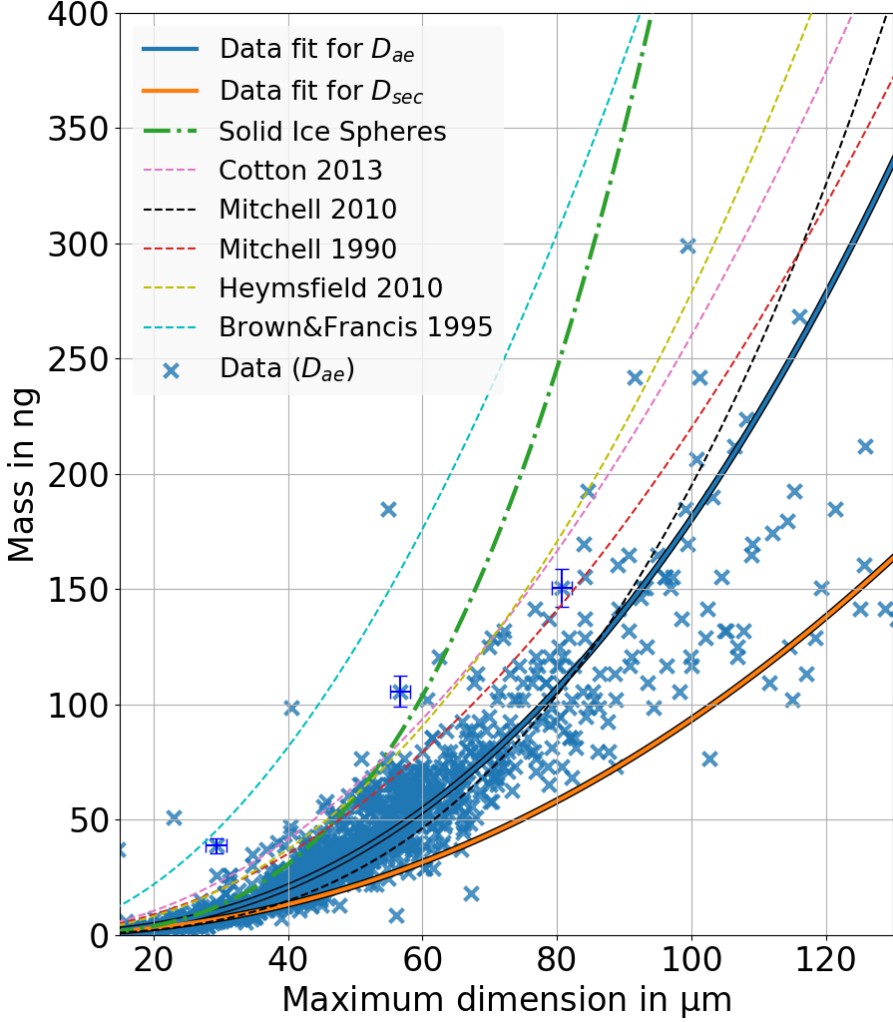

**Figure 5.** Ice crystal mass as a function of maximum dimension from the present ICC experiments (N = 1207, $D_{ae}$ in blue, best fit as solid blue line; $D_{sec}$ best fit as orange solid line). Parameterizations from literature are plotted by dashed lines. The green dash-dotted line shows the mass of a spherical object with density $\rho_{ice} = 0.9184\,\mathrm{kg\,m^{-3}}$

.

The hydrodynamic diameter of the falling crystals, which serves as a good descriptor of the hydrodynamic properties of a falling object, can be calculated from Eq. (7):

$$D_{hyd} = \frac{g}{3\pi\eta}\frac{m}{v}. \qquad (9)$$

If $D_{hyd} < D_{max}$, the observed falling object has a ratio between $m$ and $v$ that is smaller than that of a sphere of diameter $D_{max}$. In Fig. 7, $D_{hyd}$ is shown as a function of $D_{maj}$ for all crystals observed in the fall track experiments. $D_{hyd}/D_{maj} < 1$ for



260 crystals with $D_{maj} <100$ µm, increasing with $D_{maj}$ and crossing the value of 1 ($D_{hyd} = D_{maj}$) at around $D_{maj} = 100$ µm. Crystal habit and size show good correlation, as most crystals with $D_{maj} <70$ µm have grown with a columnar or irregular habit, and larger crystals were mostly dendritic or aggregated. Nevertheless, no distinct dependence of the ratio $D_{hyd}/D_{maj}$ on habit can be observed as seen in Fig. 8). The difference between the mean ratios $D_{hyd}/D_{maj}$ observed in each of the other habits is smaller than the standard deviation of $D_{hyd}/D_{maj}$ within each habit class (represented by error bars). The small

265 mean ratio for capped columns is an artifact of the small sample size of this particular habit. The relationship between $D_{hyd}$ and $D_{max}$ for crystals of all observed sizes and shapes follows the power law $D_{hyd} = 0.039D_{max}^{1.69}$.

 Additionally, a separate analysis of columnar crystals has been conducted to complement the investigation where all crystals of different habits were combined. Columns were chosen due to their abundance in the experiments (over 20% of all observed crystals) and their symmetric shape, which allows for an appropriate estimation of their projected area during fall. The Best

270 numbers $X$ (see Eq. (5)) of the observed columns ranges between $10^{-1}$ and 10 (see Fig. 9), with Reynolds numbers (see Eq. (4)) between 0.05 and 0.5. The mean aspect ratio (AR) of columns investigated in this work was 0.49. The data fit (orange line, with its uncertainty as gray shading) is enveloped by both curves from Jayaweera and Cottis (1969), who determined $X$ and $Re$ for metal cylinders with two different aspect ratios falling in motor oil. The power law relationship suggested by Bürgesser et al. (2016) generally predicts significantly higher Best numbers than we observed for a given Reynolds number.

275 For low Reynolds numbers ($Re \ll 1$), both theoretical models and experimental studies suggest that the orientations of falling columns are randomly distributed (Westbrook, 2007; Bürgesser et al., 2016). The same behavior can be seen in the distribution of orientations of the falling columnar crystals in our study, which do not show any preferred alignment of crystals to the fall direction (see Fig. 10).

### 4.4 Fall streak measurements

280 The velocity range measured using particle streaks during the validation experiment was similar to the range prevalent in the holographic measurements, with a mode in the velocity distribution around $v_{sed} = 40$ mm/s for both techniques. To further characterize the fall behavior of crystals in the fall section, the spatial distribution of fall streak center points detected in each part of the sample volume during the validation experiment is shown in Fig. 11b. Streaks could be observed throughout the entire field of view of the camera. Nevertheless, the image edges were slightly less populated, which is a result of the filtering of

285 incomplete fall streaks extending outside of the field of view. Figure 11c shows the evolution of the mean particle fall velocity over time for the fall streak experiment. The dashed lines show the moving average of the fall speeds' standard deviation in each image. The highest mean velocities were detected in the early phase of the experiment, as the fastest crystals arrived in the sample volume first. After around 10 s, the velocity reached a steady level. In this phase, a mix of crystals with high and low fall velocities was present in each layer of the chamber due to the constant resupply of newly formed crystals. From this point

290 on, a slow decline of the mean fall velocity was observed because the crystals remaining in the section slowly sedimented out.

 Figure 12 shows the distribution of velocities from a set of streak measurements (top panel) and the size distribution of ice crystals measured under the microscope afterwards. A similar general shape can be observed, with a steep increase from low values to a mode in an intermediate region (15 mm/s for $w$, 30 µm for $D_{ae}$) and a longer tail towards higher values. The bottom

panel shows the histogram of sedimentation velocities calculated from $D_{ae}$ following Stokes theory (Eq. (7)) for each crystal, using the $m(D)$ power law determined in subsection 4.2 for mass calculation ($a = 0.4972$, $b = 2.36$). While the general shapes of the distributions are roughly similar, the mode of observed velocities (top panel) is found at slightly lower velocity values than the one in the distribution predicted by Stokes theory. This implies that the observed crystals are subjected to a stronger drag force than a spherical object with diameter $D_{ae}$ falling in the Stokes regime.

## 5 Conclusions

During experiments conducted in the ice cloud chamber of the Mainz vertical wind tunnel laboratory, in-focus images of small ice crystals with sizes between 25 and 220 μm during their sedimentation in a calm environment from reconstructed holograms were produced. From these images, sedimentation velocities of over 3500 particles have been obtained by particle tracking. After classifying the crystals based on their habits, a relationship between hydrodynamic and maximum diameter was calculated. A separate analysis was conducted for columnar crystals, which were the most frequently observed crystals of regular shape. The relationship between Best and Reynolds numbers that was determined for columnar crystals agreed well with the parameterization from Jayaweera and Cottis (1969). The mass of 1207 crystals was determined by collecting the crystals on a glass slide and measuring their size before and after melting. A parameterization relating particle mass and maximum dimension was calculated, which describes the properties of ice crystals in the investigated size range more accurately than similar relationships found in the literature.

The analysis methods used for determining the particle properties were almost entirely automated requiring minimal operator interaction, owing to the capabilities of modern computer vision and machine learning algorithms. The accuracy of data obtained through these automated processes was validated through comparison to operator-labeled samples. The automation accelerated the acquisition and analysis of new data.

Sensitivity studies on the effect of the proposed mass parameterizations on atmospheric models should be conducted in order to evaluate their impact on the formation and persistence of clouds containing small ice crystals, because the processes involved include too many complex feedback mechanisms to allow for an immediate, general conclusion. Conducting such sensitivity studies is suggested here, as our literature search did not reveal any assessments investigating the subject.

**The Supplement related to this article is available online at [insert supplement URL].**

*Code and data availability.* Software code and experimental data are freely available upon request to the contact author.

*Author contributions.* MW, SKM, JF, and SB designed research and instrumentation; MW performed research and evaluated data; MW, MS and SB drafted the manuscript.





*Competing interests.* The authors declare that they have no conflict of interest.

*Acknowledgements.* This work was supported by the Max Planck Graduate Center and internal funds from the Max Planck Institute for Chemistry.



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

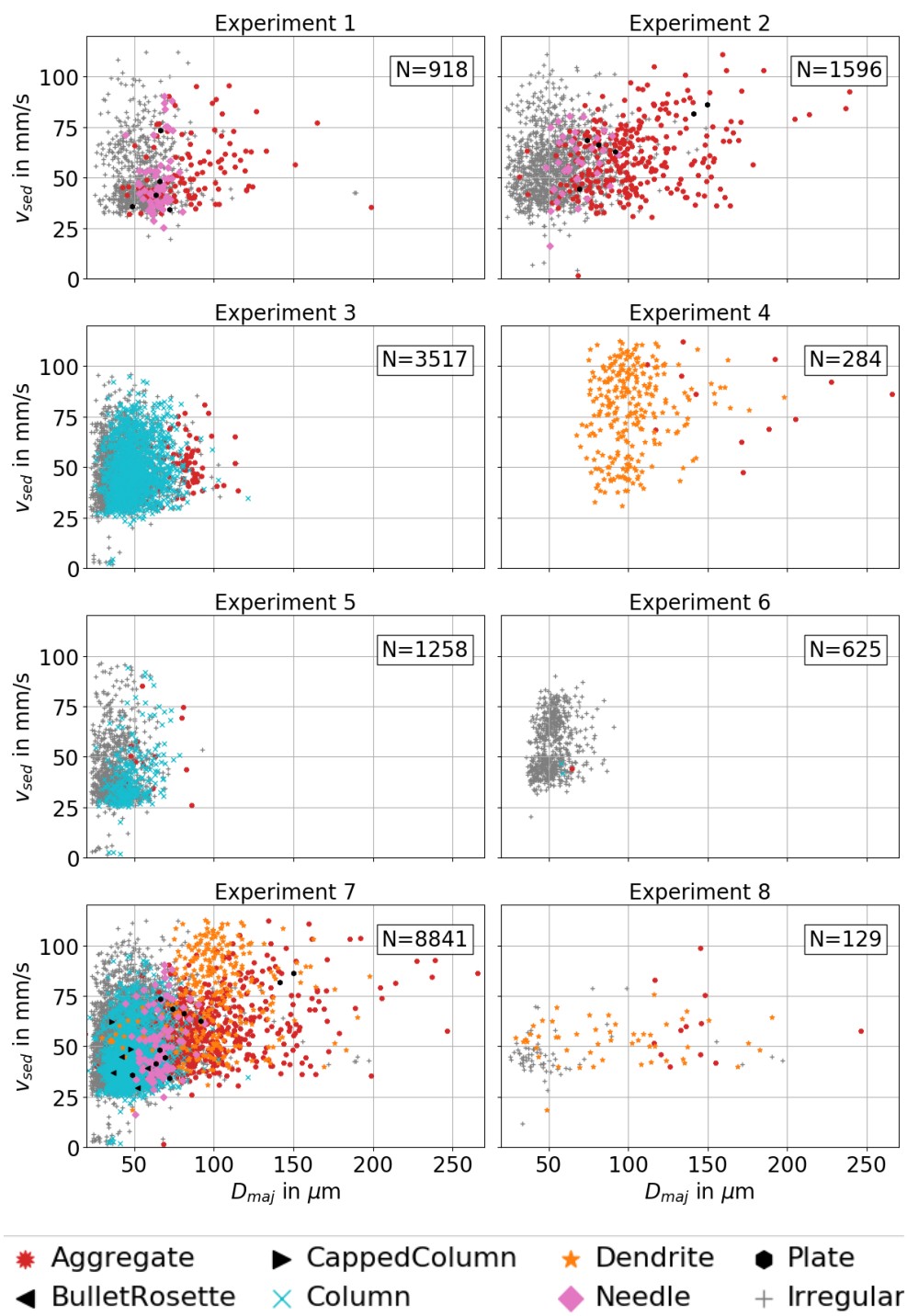

**Figure 6.** Sedimentation velocity ($v_{sed}$, in mm/s) and size ($D_{maj}$ in µm) measurements from holography particle tracking experiments (particle numbers in each experiment are annotated).





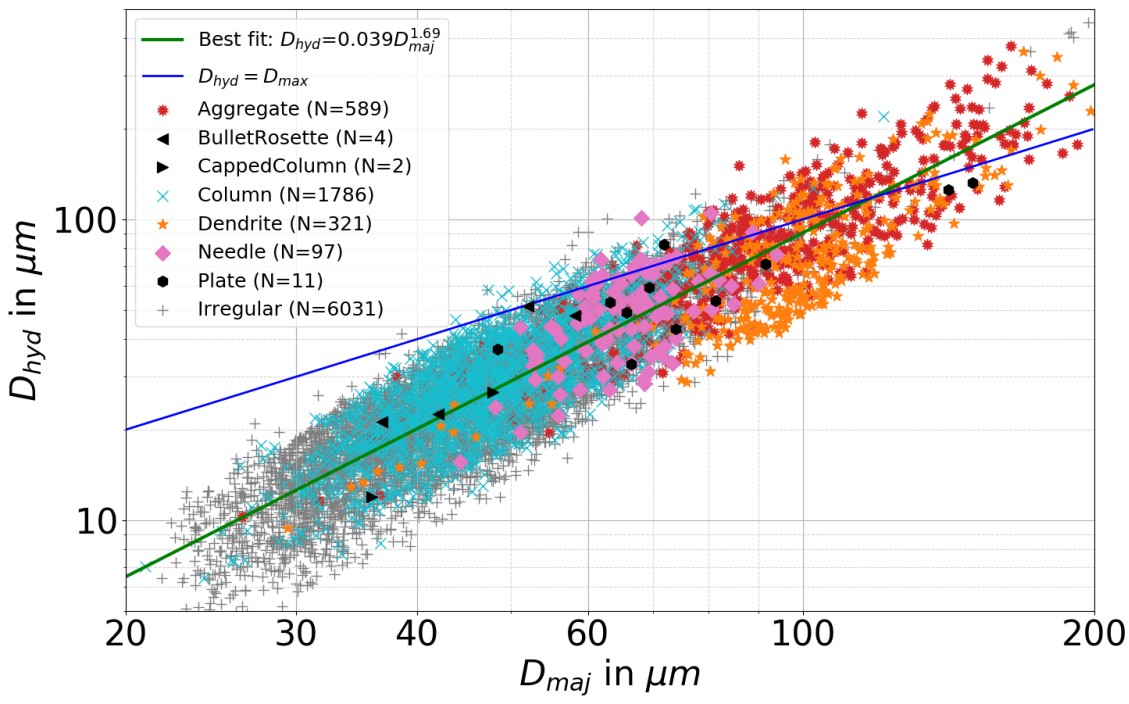

**Figure 7.** Relation between the hydrodynamic diameter $D_{hyd}$ calculated using Eq. (7) and the measured maximum dimension $D_{max}$ of falling crystals. Different crystal habits (classified by the trained predictor) are marked as different symbols and colors. Power law fit as green line, $D_{hyd} = 0.039 D_{maj}^{0.69}$. The blue line represents $D_{hyd} = D_{max}$.

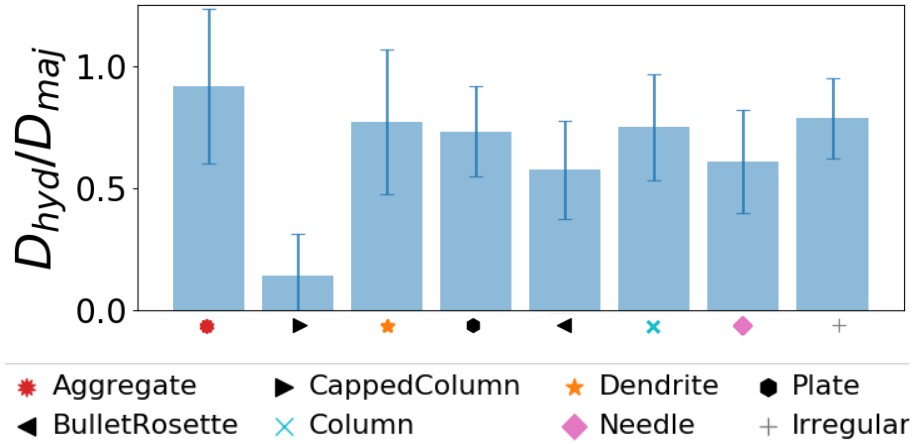

**Figure 8.** Mean value of $D_{hyd}/D_{maj}$ for each crystal habit for data shown in Fig. 7.



**Figure 9. (a)**: Best number as function of Reynolds numbers for investigated falling columnar crystals, $N = 1844$. Data fit added in orange with error range as gray shading. Parameterization from Jayaweera and Cottis (1969) in magenta and orange, from Bürgesser et al. (2016) in green. **(b)**: Aspect ratio histogram of investigated columnar crystals, mean aspect ratio $\overline{AR} = 0.49$.

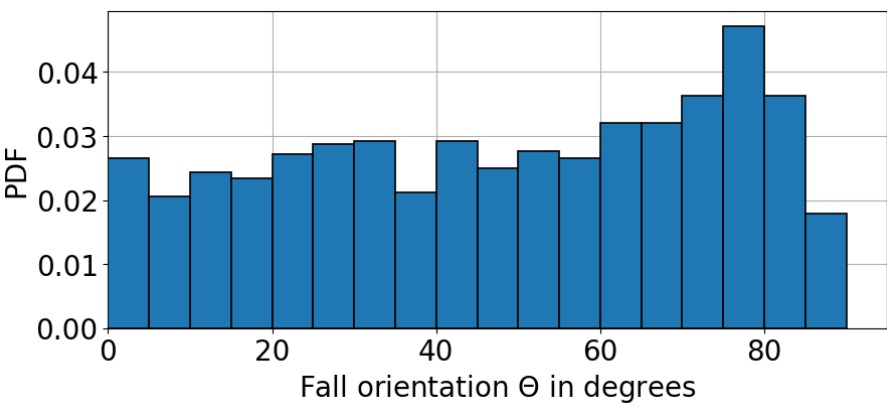

**Figure 10.** Histogram of the falling columnar crystals' orientation Θ, with 90 ° corresponding to a fall with the major axis normal to the falling direction. $N = 1844$.

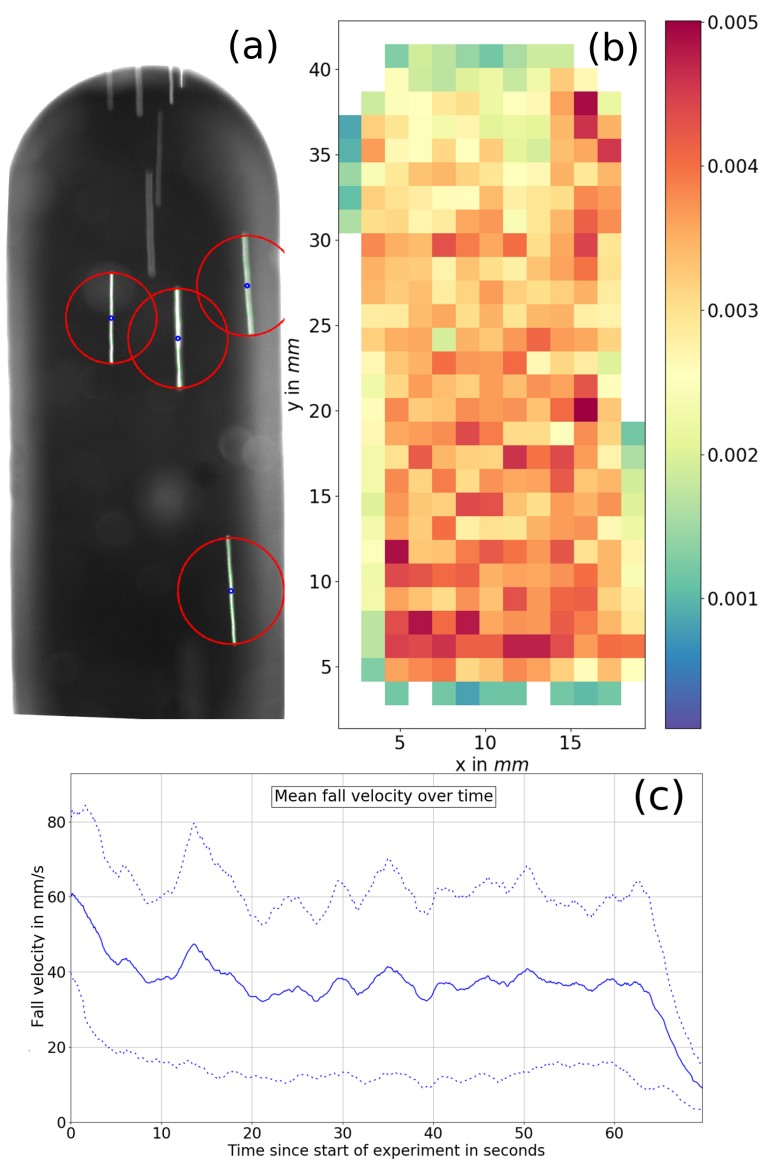

**Figure 11. (a)**: Example fall streak image. Detected streaks are circled in red, diameter is equal to the detected streak length. **(b)**: Relative occurrence of crystals in partial regions of the sample volume, number of streak center points observed in pixel region divided by number of streaks in the full image. Total number of streaks in experiment: $N = 24775$. **(c)**: Mean velocity of all falling objects in the sample volume over time. A moving average over 2 seconds was used to smooth the time series. The dashed lines show the moving average of the fall speeds' standard deviation in each image.

**Figure 12.** Distribution of different parameters for fall streak experiment. **(a)**: crystal fall velocities extracted from streak images. **(b)**: area-equivalent diameter ($D_{ae}$) from microscope images of collected crystals. **(c)**: fall velocity predicted from $D_{ae}$ size distribution using Stokes theory (Eq. (7)).