# Peer review of "Technical information: Object detection details"

_Atmospheric Chemistry and Physics, 2020_

## Referee Comment (RC1) · Andrew Heymsfield (Referee) · 7 Jun 2020

Review of "Application of holography and automated image processing for laboratory experiments on mass and fall speed of small cloud ice crystals", by Weitzel et al., submitted to ACP.

This is an interesting article that uses the small chamber at the University of Mainz to grow small ice crystals and then measures their sizes, masses, terminal velocity and orientation and velocity, Reynolds Number and Best Number, during fall. The proper-

ties of small ice crystals, notably mass and terminal velocity, and quite poorly known and the goal of this article is to develop a better understanding of these properties.

There are some novel methods used in this study. Most notably, the use of a holographic camera to determine their properties, and of course the Mainz tunnel. I have done work directly related to this article and have gone through the article very carefully.

There's a major difficulty that I've identified and have suggestions for how this can be amended in a future article. This has to do with the representativeness of the results for natural clouds. Here's the problem. Ice crystals are generated with a liquid nitrogen cooled rod, inserted into the chamber. The rod has a temperature (close to that of the liquid nitrogen) of below -1950C, according to the article. This homogeneously nucleated the droplets. First, the vast majority of the ice crystals take on the shape of the ice crystals that are nucleated at a temperature of -300C or below rather than the ambient temperatures of their later growth (-8 to -160C). This can readily be seen in Figures 2 and 4, and as noted on lines 237-238 "the majority of ice crystals ($\sim$68%) showed irregular crystal growth, and aggregates. At these temperatures and for the sizes considered (equivalent diameters between 15 and 145 microns) the crystals should be dominantly planar and dendritic, not the types observed, especially aggregates, which generally form at sizes above 200 microns, suggesting very high ice concentrations. Second, the ice crystals grown are not representative of the temperatures of the temperatures used in the study, and are very unlike ice crystals that are growth in natural clouds. This was a feasibility study; perhaps in a future study ice crystals representative of the growth of natural crystals could be studied.

A related study, by Ryan et al. (1976, JAS), "The growth rates and densities of ice crystals between -3 and -210C" , covered the size range and temperatures used in the Weitzel et al study. Cloud droplet sizes in their cloud chamber were between 4 and 30 microns. Ice crystals were produced using a small pin cooled in liquid nitrogen and even with this procedure concentrations of 5-20 cm-3 were produced. Even so,

the resulting ice crystal habits were entirely consistent with experiments that grew ice crystals on silk filaments, not the types that were observed in this study. Indeed, they note that "in our experiments, more complicated structures, such as polycrystalline crystals, dendrites and capped columns were never observed". They measured ice crystal masses, densities, and axial dimensions. Although they did not have the very large sample size of the crystals in this study, they do provide very useful results.

As noted above, my suggestion is that in a future study, the authors devise a method to nucleate a few crystals at a time, possibly with ice nuclei in a cage within the chamber, then open the cage and let the particle grow at their correct temperature and ice supersaturation.

I have several minor comments but wanted to get this review to ACP for discussion.

Andy Heymsfield NCAR June 3, 2020

---

## Referee Comment (RC2) · Anonymous Referee #2 · 23 Jun 2020

This is a useful experimental study which presents new and valuable data on the fall speeds and orientations of small ice crystals. I think this should be published, but I feel the paper would benefit from a number of minor corrections and modifications. See annotated PDF for details.

Please also note the supplement to this comment:
https://www.atmos-chem-phys-discuss.net/acp-2020-339/acp-2020-339-RC2-supplement.pdf

---

## Referee Comment (RC3) · Anonymous Referee #3 · 29 Jun 2020

This manuscript presents the results of a series of experiments designed to measure the size, mass, and fall velocity of small ice crystals. Focusing on particles smaller than 150um in diameter, this study fills a large gap in the literature where detailed measurements of the physical properties of small ice are rare, and has important followup implications in cloud lifetimes, radiative properties, and cloud dynamics. Overall I think this manuscript is well written, with clearly described techniques, methods, and results. More discussion of the results in a few areas of the manuscript would be valuable, as well as some points of clarification as noted below. Otherwise, I have no

major reservations with the work presented, and recommend publication in ACP with minor revisions.

Line 85: Were particles measured in the fall chamber individually matched to the particles collected on the glass slides? It was not clear if the experiment supported this.

Line 100: What is the estimated positional accuracy in all three directions for particles in the hologram?

Line 132: How many particles are in a typical hologram? Was the ice concentration so high that linking particles from one frame to another is difficult?

Line 180: Were the same edge detection methods used for the holographic images as for the slide-captured images?

Line 242 and Figure 5: Were the other power law relationships converted to use a consistent size definition (Deq or Dsec)? This can sometimes make a large difference.

Line 243: What are the power law coefficients from this study (a and b), for both Deq and Dsec?

Line 245: It is mentioned in the abstract that the other power laws were generally developed on larger particles and have been extrapolated down to the sizes in this study. I think this point needs to be reemphasized here. Some discussion behind the observed differences would also be valuable, such as the types of particles (habit, degree of riming, etc.) that were collected in the other studies. Also, is there a functional form that could bridge the gap between various small/large mass-size parameterizations?

Line 255: Related to the first comment, is the mass of each particle known, i.e. were the velocity measurements (either by hologram or fallstreak) directly linked to the mass measurement for each individual particle? If not, is mestimated from the power law in Section 4.2 to get Dhyd?

Line 260: What is happening physically when Dhyd > Dmax, and do you have any

speculations or measurements to indicate why that transition occurs around 100um?

Line 278: The 3-D holographic track information is highlighted in the abstract and in a few places in the body of the manuscript, but I don't see any data on the lateral movement of the particles presented in this manuscript. Is there significant lateral movement of the particles? Were any tumbling motions observed? I think it would be valuable to add a figure or two to highlight any lateral movement (or lack thereof).

Line 295: Was there any attempt to measure the size of the particles in the fallstreak analysis, and how does the distribution compare with the holographic method?

СЗ

---

## Author Comment (AC1) · 19 Aug 2020

First of all, we would like to thank the reviewer for the useful comments and suggestions which helped to improve the manuscript. The reviewer's comment were answered in the following.

The homogeneous nucleation method was developed initially to confirm the feasibility of the chamber experiments. It is a simple approach that does not introduce pollution with aerosol particles into the chamber and provides a reliable source of nucleation of

a substantial amount of crystals. That said, a smaller number of experiments using INP as nucleation source were conducted to form an understanding of the influence of the nucleation mechanism on the crystals' growth behavior. While no quantitative analysis on these observations was conducted, a similar fraction of crystals with irregular shapes were observed after heterogeneous nucleation induced by ice nuclei (Montmorillonite). This suggests that the observed crystal shapes are influenced by the thermodynamic conditions during particle growth within the chamber rather than by the conditions during initial freezing. As the reviewer suggests in their comment, a more thorough analysis of the differences between particle growth after homogeneous and heterogeneous nucleation in the chamber would be interesting, and can be subject of future work. However, we do not expect any influence of the freezing mode (i.e. homogeneous or heterogeneous) on the fall velocity of ice crystals of the same habit. An elaboration on the uncertainties in the understanding of the growth processes within the chamber has been added to the discussion section of the article (Line 222 and the following).

---

## Author Response (AR1)

**Manuscript entitled „Application of holography and automated image processing for laboratory experiments on mass and fall speed of small cloud ice crystals" by M. Weitzel et al.**

We would like to thank all reviewers for the useful comments and interesting suggestions which helped improve the manuscript. The reviewers' comments and questions are answered in the following.

Remark:
The reviewers' comments are written in bold font, our answers in standard font.

**AUTHOR'S REPLIES TO REVIEWERS' COMMENTS**

**Authors' response to reviewer #1:**

" […] First, the vast majority of the ice crystals take on the shape of the ice crystals that are nucleated at a temperature of -30 °C or below rather than the ambient temperatures of their later growth (-8 to -16 °C). This can readily be seen in Figures 2 and 4, and as noted on lines 237-238 "the majority of ice crystals (~68%) showed irregular crystal growth, and aggregates. […] At these temperatures and for the sizes considered (equivalent diameters between 15 and 145 microns) the crystals should be dominantly planar and dendritic, not the types observed, especially aggregates, which generally form at sizes above 200 microns, suggesting very high ice concentrations."

The homogeneous nucleation method was developed initially to confirm the feasibility of the chamber experiments. It is a simple approach that does not introduce pollution with aerosol particles into the chamber and provides a reliable source of nucleation of a substantial amount of crystals. That said, a smaller number of experiments using INP as nucleation source were conducted to form an understanding of the influence of the nucleation mechanism on the crystals' growth behavior. While no quantitative analysis on these observations was conducted, a similar fraction of crystals with irregular shapes were observed after heterogeneous nucleation induced by ice nuclei (Montmorillonite). This suggests that the observed crystal shapes are influenced by the thermodynamic conditions during particle growth within the chamber rather than by the conditions during initial freezing.

As the reviewer suggests in their comment, a more thorough analysis of the differences

between particle growth after homogeneous and heterogeneous nucleation in the chamber would be interesting, and can be subject of future work. However, we do not expect any influence of the freezing mode (i.e. homogeneous or heterogeneous) on the fall velocity of ice crystals of the same habit.

An elaboration on the uncertainties in the understanding of the growth processes within the chamber has been added to the discussion section of the article (Line 222 and the following).

**Authors' response to reviewer #2:**

**"There are other ´experimental´ studies of velocity of small ice particles in chambers e.g. those cited in Westbrook paper you mention (which are quite old) but also the more recent work by Argentinian group […] It would be very useful to put you work in context against those other studies"**

In their work, the Argentinian group Bürgesser et al. studied the fall behavior of hexagonal planar (Bürgesser & Castellano, 2017) and column crystals (Bürgesser et al. 2016) by determining and relating Best- and Reynolds numbers. We have compared our results to the results of the latter study in Section 4.3 and Figure 9. This comparison indicated that the parameterization of Bürgesser et al. predicts significantly higher Best numbers than our experimental results.

As our experiments did not include many hexagonal planar crystals, we were not able to accurately establish whether our observations agreed or disagreed with the findings in Bürgesser & Castellano (2017).

**"It might also be cold enough to nucleate ice homogeneously from vapour if there is flow of warmer air past this rod?"**

Because of the large number concentration of ice crystals homogenously nucleated from droplets present in the air flow, we expect the pathway of homogenous nucleation to be negligible in this setup. The copper rod is kept at this very low temperature only for a limited time before it is flushed with warmer air. During this limited time of cold finger activation, the droplet supply is not expected to be depleted, thus crystals nucleated from droplets are always present. Supersaturations high enough to trigger homogenous nucleation from water vapour are thus very unlikely to be reached due to the preferred pathway of diffusion of water vapour towards the preexisting crystals.

**"You could be more specific here. Do the points [Fig. 3b] outside the grey line tell us about the accuracy of the velocity estimates?"**

The uncertainty of the velocity measurements themselves is significantly smaller than the spread observed here. The spread is thus a result of a superposition of the residual turbulence in the chamber and the accuracy of the velocity estimates.

**"Are you able to match mass and diameter estimates from the particles on the slide to the corresponding velocity measurements of the particles in the tube? Or are you characterizing the average mass of similar crystals at around the same time (and what random error does that introduce?)**

The concept of matching individual velocity measurements to mass measurements was considered during design of the experimental setup. The final setup, however, only allows for the comparison of ensembles of mass measurements to ensembles of velocity measurements, as the focus of this study was to maximize the number of individual m(D) and v(D) data points. The connection between our findings for mass and velocity can only be made by comparing the distributions of particle masses and the distribution of fall velocities measured during the same experiment.

This fact has been reemphasized in the corresponding text section of the revised manuscript in Section 3.2, lines 177 and following.

It shall be noted that this response has been repeated for the answer to a comment given by reviewer #3.

**"I think it would be very useful to estimate the Reynolds number of the particles somewhere. Then you can establish the extent to which we should expect to be in the Stokes regime, as a function of D" […]**

The Reynolds number of the observed crystals ranged between 0.1 and 0.7 (see Fig. 9). The observed fall behavior is thus not clearly in the Stokes regime (where Re would be << 1), with turbulence showing a minor impact on the observed fall velocity. A note has been added in line 307 and the following in the revised manuscript.

**"In the Stokes regime it is surely not possible for $D_{hyd} > D_{max}$, but $D_{maj}$ is not equal to the maximum span I suppose (how does it relate?) and perhaps at $D_{maj} = 100$ microns are you moving out of the Stokes regime?"**

The description of ice crystals with respect to their maximum span $D_{max}$ was never intended in this context, and the text and figures have been adjusted accordingly to correctly always use $D_{maj}$.

$D_{hyd}$ is calculated from Equation 9. The mass power law relation given in Section 4.2 is applied to parameterize m on the right side of the equation here, which introduces two sources of error. Firstly, the parameterization is determined for the area-equivalent diameter of the crystal contour, $D_{ae}$, but applied to the long axis of an ellipse fit around the particle contour $D_{maj}$ and thus not applicable strictly without error in this context. Further, the mass parameterization is most strongly determined by ice particles with sizes around 60 µm and, as evident from Fig. 5, mostly overestimates the mass of crystals with D > 100 µm. This overestimation of m also leads to an overestimation of $D_{hyd}$ for those larger crystals. We thus do not expect that $D_{hyd} > D_{maj}$ would be observed for any crystals in individual measurements, but rather an asymptotical approximation of the fit to $D_{hyd} = D_{maj}$.

The relationship between $D_{hyd}$ and $D_{maj}$ proposed in this work is thus expected to accurately describe crystals with $D_{maj} < 90$ µm. For larger crystal sizes, more data would be required to either determine a new parameterization or adjust the one given here to be more accurate for all $D_{maj}$.

**"Meaning is not clear – what is correlated with what, and how?"** *(Re: Line 261 in the original draft, "Crystal habit and size show good correlation, as most crystals with $D_{maj} < 70$ µm have grown with a columnar or irregular habit, […]")*

We observed that none or very few of the observed columnar or irregular crystals grew to sizes $D_{maj} > 80$ µm. Most of the observed crystals between 80 and 200 µm were of dendritic shape or aggregates crystals, which is why we concluded a correlation between crystal size and habit in our observations. The text section in line 287 has been adjusted to be clearer.

**"Re-X relationships: the authors could go a lot further in analysing the accuracy of parameterisations/theories, e.g. Mitchell 1996, Böhm 1989/1992, and Heymsfield and Westbrook 2010. You seem to have all the data to do this. Why not?"**

We focused on the development and description of a new measurement technique for determining the properties of ice crystals in the size range smaller than 150 µm. While comparisons to related studies are important for understanding the context of this work's results, and have thus been conducted and described in Section 4, further analysis of the accuracy of other parameterizations is beyond the scope of this paper.

**"I think the paper would greatly benefit from (i) more consistent use of characteristic length scale throughout, (ii) more explanation in the text of the rationale for picking a particular D for a given analysis or plot."**

We incorrectly mentioned the particle maximum diameter $D_{max}$ as the considered characteristic length scale in Figure 7 and the following text. These errors have been corrected, as $D_{maj}$ is used for every aspect of this work where the results from holographic particle tracking are discussed.

To further clarify the usage of different characteristic length scales, Section 3.3 has been expanded to explain more clearly which formulation is used where and why.

**"Consider providing data as a table / text file, as supplementary material?**

We chose not to directly attach our data to this publication, as the data set is too large for convenient viewing. A publication on a suitable platform is planned at a future date. Nevertheless, all requests for our experimental data are very welcome and we are going to provide them for further scientific use.

**Authors' response to reviewer #3:**

**"Line 85: Were particles measured in the fall chamber individually matched to the particles collected on the glass slides? It was not clear if the experiment supported this"**

Reviewer #2 highlighted a similar point, and the same explanation given to them is appropriate here.

The concept of matching individual velocity measurements to mass measurements was considered during design of the experimental setup. The final setup, however, only allows for the comparison of ensembles of mass measurements to ensembles velocity measurements, as the focus of this study was to maximize the number of individual m(D) and v(D) data points. The connection between our findings for mass and velocity can only be made by comparing the distributions of particle masses and the distribution of fall velocities measured during the same experiment.

This fact has been reemphasized in the corresponding text section of the revised manuscript in Section 3.2, lines 180 and following.

**Line 100: What is the estimated positional accuracy in all three directions for particles in the hologram?**

The estimated positional uncertainty is $\Delta x = \Delta y = 9$ μm, and $\Delta z = 200$ μm along the optical axis. Since the fall velocity is calculated as the vertical component of the particles' motion, only $\Delta y$ matters for the uncertainty in w. The resulting velocity uncertainty of $\Delta w_{track} = 0.5$ mm s$^{-1}$ is considered along with the error introduced by residual turbulence (see Section 3.1.4), and a text section has been added in line 111 and the following to elaborate on this aspect.

**Line 132: How many particles are in a typical hologram? Was the ice concentration so high that linking particles from one frame to another is difficult?**

The most populated holograms contained up to several hundred crystals, which made linking challenging. The median particle number, however, was around 20. For these typical holograms, the third dimension made linking mostly easy.

**Line 180: Were the same edge detection methods used for the holographic images as for the slide-captured images?**

The detection method for hologram analysis involved a thresholding algorithm in the reconstructed slices. As many two-dimensional slices are reconstructed for each hologram, the signal created by the crystals are visible in several layers along the optical axis. This three-dimensional nature of the detected signal makes particle detection and noise filtering easier than in the case of classical two-dimensional imaging.

In the 2D case of microscope image analysis, thresholding often introduces errors created by incomplete edges or incorrect merging of multiple objects. To improve measurement accuracy, the more sophisticated segmentation methods described in Section 3.2 and the Supplement were implemented and compared.

**Line 242 and Figure 5: Were the other power law relationships converted to use a consistent size definition ($D_{eq}$ or $D_{sec}$)? This can sometimes make a large difference.**

We agree that the size definition has to be taken into account when applying power law relationships to determine the unknown mass of an ice particle from its size. The definition used plays a major role regarding the applicability and accuracy of parameterizations like the ones determined in this work. The parameterizations depicted in Figure 5 are shown without prior conversion however, as either additional information about the particles at hand or simplifying assumptions would have been required for an accurate conversion in several cases.

**Line 243: What are the power law coefficients from this study (a and b), for both $D_{eq}$ and $D_{sec}$?**

$D_{sec}$: a = 0.03097, b = 2.13
$D_{ae}$: a = 0.4972, b = 2.36

The power law relationships including their parameters were added in Section 4.2, line 257 and the following.

**Line 245: It is mentioned in the abstract that the other power laws were generally developed on larger particles and have been extrapolated down to the sizes in this study. I think this point needs to be reemphasized here.**

A remark has been added in line 279 which reemphasizes that the power law relationships from the literature were determined from measurements of larger ice crystals.

**Some discussion behind the observed differences would also be valuable, such as the types of particles (habit, degree of riming, etc.) that were collected in the other studies.**

Section 4.2 has been expanded by more detailed descriptions of the origins of the parameterizations shown in Figure 5.

**Also, is there a functional form that could bridge the gap between various small/large mass-size parameterizations?**

We have not looked into determining a functional form to bridge the gap between several parameterizations. A future review article with the objective of finding such a relationship would be a valuable resource for handling the challenges of parameterizing particle mass.

**Line 255: Related to the first comment, is the mass of each particle known, i.e. were the velocity measurements (either by hologram or fallstreak) directly linked to the mass measurement for each individual particle? If not, is m estimated from the power law in Section 4.2 to get $D_{hyd}$?**

This question is mostly answered in our response on the first comment by the reviewer.

m is indeed parameterized from the power law obtained in Section 4.2 for the calculation of $D_{hyd}$. The errors introduced thereby are discussed in line 298 and the following.

**Line 260: What is happening physically when $D_{hyd} > D_{max}$, and do you have any speculations or measurements to indicate why that transition occurs around 100um?**

Again, Reviewer #2 asked a similar question, and the explanation given to them is repeated here.

$D_{hyd}$ is calculated from Equation 9. The power law relation given in Section 4.2 is applied to parameterize m on the right side of the equation here, which introduces two sources of error. Firstly, the parameterization is determined for the area-equivalent diameter of the crystal contour, $D_{ae}$, but applied to the long axis of an ellipse fit around the particle contour $D_{maj}$ and is thus not applicable strictly without erroring this context. Further, the mass parameterization is most strongly determined by ice particles with sizes around 60 µm and, as evident from Fig. 5, mostly overestimates the mass of crystals with D > 100 µm. This overestimation of m also leads to an overestimation of $D_{hyd}$ for those larger crystals. We do

not expect that $D_{hyd} > D_{maj}$ would be observed for any crystals in individual measurements, but rather an asymptotical approximation of the fit to $D_{hyd} = D_{maj}$.

The parameterization between $D_{hyd}$ and $D_{maj}$ proposed in this work is thus expected to accurately describe crystals with $D_{maj} < 90$ µm. For larger crystal sizes, more data would be required to either determine a new parameterization or adjust the one given here to be more accurate for all $D_{maj}$.

The discussion section has been extended in line 297 and the following by a paragraph explaining these considerations.

**Line 278: The 3-D holographic track information is highlighted in the abstract and in a few places in the body of the manuscript, but I don't see any data on the lateral movement of the particles presented in this manuscript. Is there significant lateral movement of the particles? Were any tumbling motions observed? I think it would be valuable to add a figure or two to highlight any lateral movement (or lack thereof).**

The observed lateral distances covered by the falling particles on their short way through the sample volume was mostly small when compared to the vertical movement. A sample figure showing the three-dimensional track of a falling particle along with the evolution of its measured properties has been added to Section S6 in the supplement. The ratio between lateral and vertical movement of the majority of all sampled crystals is in a similar range.

**Line 295: Was there any attempt to measure the size of the particles in the fallstreak analysis, and how does the distribution compare with the holographic method?**

The size of particles sampled in the fall streak method was not investigated directly from the streak images. The method was designed with the objective of optimizing the accuracy and quantity of the velocity measurements, which resulted in large errors if particle size were extracted from the width of the streaks. Size and velocity observed in this method can thus only be related through the distribution of the fall velocity (determined from streaks) and size (determined from microscopy afterwards) of ensembles of many crystals.

**Manuscript entitled „Application of holography and automated image processing for laboratory experiments on mass and fall speed of small cloud ice crystals" by M. Weitzel et al.**

**List of changes in the revised manuscript**

- **Section 3.1.1, Line 107 and following:**
  o Estimation of uncertainty in particle position added.

- **Section 3.1.2, Line 136 and 137:**
  o A reference to a visualization of the information gained from a sample particle track in the Supplement was added.

- **Section 3.2, Line 180 and following:**
  o Added a paragraph explaining that and why no direct matching between mass and velocity points is possible.

- **Section 3.3, Line 214 and following:**
  o Added further elaboration about the reasoning of used characteristic length scales.

- **Section 4, Line 226 and following:**
  o Added a paragraph discussing the influence of the cold finger nucleation method on the observed ice crystal properties.

- **Section 4.2, Line 261 and following:**
  o Added parameters a and b for the m(D) power law relation.

- **Section 4.2, Line 266 and following:**
  o Added a paragraph discussing the properties of ice crystals investigated in the studies from which m(D) parameterizations are shown in Fig. 5. The sample methods are described, along with the types of crystals, degrees of riming (if applicable) and the characteristic length scale used.

- **Section 4.2, Line 279 and 280:**
  o Added a remark underlining that, except for Mitchell 2010, all studies focused on investigating ice crystals larger than 100 µm

- **Section 4.3 and Figure 7:**
  - Changed incorrectly used characteristic size $D_{max}$ to $D_{maj}$ in several occurrences in the text, figure and caption.

- **Section 4.3, Line 290 and following:**
  - Adjusted this section for clarification.

- **Section 4.3, Line 296 and following:**
  - Added a paragraph explaining the occurrence of data points with $D_{hyd} > D_{maj}$ in Fig. 7 and error estimation. Added a remark limiting the applicability of the obtained $D_{hyd}$ ($D_{maj}$) relationship to $D_{maj} < 90$ µm.

- **Section 4.3, Line 309 and 310:**
  - Added a comment about the applicability of the Stokes approximation.

- **Supplement:**
  - Added Section S6 with sample images of a typical holographic particle tracking observation and a fall streak recording

[revised manuscript text omitted]